# Temporal loudness weights: Primacy effects, loudness dominance and their interaction

**Alexander Fischenich**[1]*, **Jan Hots**[2], **Jesko Verhey**[2], **Julia Guldan**[1], **Daniel Oberfeld**[1]

**1** Department of Psychology, Johannes Gutenberg-Universität Mainz, Mainz, Germany, **2** Department of Experimental Audiology, Otto von Guericke University Magdeburg, Magdeburg, Germany

* alexander.fischenich@uni-mainz.de

## Abstract

Loudness judgments of sounds varying in level across time show a non-uniform temporal weighting, with increased weights assigned to the beginning of the sound (primacy effect). In addition, higher weights are observed for temporal components that are higher in level than the remaining components (loudness dominance). In three experiments, sounds consisting of 100- or 475-ms Gaussian wideband noise segments with random level variations were presented and either none, the first, or a central temporal segment was amplified or attenuated. In Experiment 1, the sounds consisted of four 100-ms segments that were separated by 500-ms gaps. Previous experiments did not show a primacy effect in such a condition. In Experiment 2, four- or ten-100-ms-segment sounds without gaps between the segments were presented to examine the interaction between the primacy effect and level dominance. As expected, for the sounds with segments separated by gaps, no primacy effect was observed, but weights on amplified segments were increased and weights on attenuated segments were decreased. For the sounds with contiguous segments, a primacy effect as well as effects of relative level (similar to those in Experiment 1) were found. For attenuation, the data indicated no substantial interaction between the primacy effect and loudness dominance, whereas for amplification an interaction was present. In Experiment 3, sounds consisting of either four contiguous 100-ms or 475-ms segments, or four 100-ms segments separated by 500-ms gaps were presented. Effects of relative level were more pronounced for the contiguous sounds. Across all three experiments, the effects of relative level were more pronounced for attenuation. In addition, the effects of relative level showed a dependence on the position of the change in level, with opposite direction for attenuation compared to amplification. Some of the results are in accordance with explanations based on masking effects on auditory intensity resolution.

**Data Availability Statement:** All relevant data are available on OSF: https://osf.io/kjgfp/.

**Funding:** This work was supported by a grant from Deutsche Forschungsgemeinschaft (DFG; www.

## Introduction

Loudness is one of the fundamental qualities of auditory perception. Recent research on loudness focuses on time-varying sounds. Powerful models for the prediction of loudness for stationary sounds have been proposed [1,2]. However, current models for the loudness of time-

dfg.de) to Daniel Oberfeld (OB 346/6-1) and Jesko Verhey (VE 373/2-1). The funders had no role in study design, data collection and analysis, decision to publish, or preparation of the manuscript. No additional external funding received.

**Competing interests:** The authors have declared that no competing interests exist.

varying sounds fail to include some of the more recent findings of studies on time-varying sounds like the non-uniform weighting of temporal portions of a sound. As recent research has shown (e.g., [3–8]), the first segments of a sound are of greater importance for the judgement of loudness than later parts. This *primacy effect* is well described by an exponential decay function with a time-constant of about 200–300 ms [9], where the temporal weight at the beginning of the sound is four to five times higher than the asymptotic weight, and where the weight assigned to a temporal portion of a sound is the integral of this function over the segment duration [9]. The effect occurs in the presence of background noise and is not altered by the absolute sound pressure level of the target sound [10]. The primacy effect is not predicted by dynamic loudness models [10].

Another effect observed in perceptual weights in loudness judgements is *loudness dominance*, which describes the phenomenon that parts of a sound that are louder than the other parts of the sound receive increased weights, and thus tend to "dominate" the global loudness judgement of a sound (e.g., [3,11–17]). It has been repeatedly suggested that loudness dominance effects might be related to forward and backward masking effects on the intensity resolution (e.g., [8,15,18]). The ability to judge the intensity of a target tone is impaired when the target follows shortly after a masker higher in level than the target (e.g., [19,20]). Therefore, temporal portions of a sound that are relatively higher in level compared to other temporal parts of the stimulus might reduce the intensity resolution for the remaining parts of the sound. When the intensity resolution is reduced, those parts are less informative for the actual task of loudness judgment. If now the listeners follow an "ideal observer" strategy of placing higher weights on temporal portions of a sound for which the intensity resolution is high [15,21], this would result in high weights assigned to parts of the sound that are higher in mean level and thus more informative. In a recent study [8], it has also been suggested that non-simultaneous masking effects on the intensity resolution might as well play a role in the occurrence of the primacy effect. Note that here and in the following, forward (and backward) masking refers to non-simultaneous masking in intensity resolution and not in signal detection. For a level-fluctuating sound, the probability that a given temporal segment is forward-masked by a temporal segment higher in level increases with the number of segments that precede the given segment. This is because when the segment levels are drawn independently from a random distribution with the same mean, the probability that a given segment is preceded by a segment that is, e.g., 10 dB higher in level, is lower for the first few segments than for later segments. As a consequence, on average the intensity resolution for later temporal parts of a sound might be reduced. This would result in a primacy effect if listeners used a strategy of assigning higher weights to temporal portions of a sound for which the intensity resolution is high.

In the present study, three experiments were conducted in order to gain a more detailed insight into the effects of the relative level of temporal stimulus components on perceptual weights in a loudness judgement task. In Experiment 1, the effect of amplifying or attenuating certain temporal segments of a sound on the loudness weights was measured for sounds consisting of four 100-ms wideband noise segments separated by 500-ms silent gaps. For this type of sounds, we expected virtually no primacy effect, because there is evidence that the mechanism causing the primacy effect recovers during silent gaps [8]. Also, forward and backward masking effects on intensity resolution are relatively weak at inter-stimulus intervals of 500 ms [19,22]. Thus, if the dominance effect is caused by such non-simultaneous masking effects, one would expect no dominance effect for this kind of sound. We varied the mean level of the first and the third segment, in order to investigate whether the position of the level change affects the size of the potential loudness dominance effect in the absence of a primacy effect. The segment deviating in mean level from the remaining segments was either amplified or

attenuated, and the amount of attenuation and amplification was varied. This was done in order to investigate if the potential effect of amplification and attenuation is *symmetric* in the absence of the primacy effect. We use the term "symmetric" in the sense that, for instance, an increase in mean segment level by 10 dB results in an increase in the segment weight by the same factor (e.g., the weight increases by a factor of 3.0) as the decrease in the segment weight caused by a reduction in segment level by 10 dB (e.g., the weight is reduced by a factor of 3.0).

In Experiment 2, we investigated whether in conditions where a primacy effect is observed in the pattern of temporal weights, the effects of amplification and attenuation and of the position of the amplified/attenuated segment within the sound deviate from the pattern observed for conditions without a primacy effect (Experiment 1). Put differently, we were interested in whether the dominance effects are modulated by the presence of a primacy effect, which would indicate an interaction between the two effects. Thus, in Experiment 2, the temporal segments were presented without silent gaps, and thus a primacy effect was expected [8]. Like in Experiment 1, the segment deviating in mean level from the remaining segments was either amplified or attenuated and we varied the position of the level change to assess whether the dominance effect differs between the different positions.

Furthermore, the comparison between the effects of relative level observed in Experiment 1 and Experiment 2 could give insight into whether or not non-simultaneous masking effects on the intensity resolution play a role for the dominance effect. In Experiment 2, such masking effects were likely to occur, because the segments were presented contiguously. In contrast in Experiment 1, segments were separated by 500 ms gaps, which should drastically reduce masking effect on the intensity resolution [22]. If masking effects play a role in the dominance effect, the size of the dominance effect in Experiment 2 should be larger than that observed in Experiment 1.

In Experiment 3, we presented sounds consisting of four 100-ms segments separated by 500 ms gaps, as well as sounds consisting of four contiguous segments of 100-ms and 475-ms duration. We presented those three types of sounds to investigate whether potential differences between the effects of loudness dominance in Experiment 1 and 2 can be attributed to the gaps between the segments or the total duration of the sounds. Furthermore, we wanted to confirm the between-subjects effects of the silent inter-segment gaps observed in Experiment 1 (with gaps) and 2 (without gaps) in a within-subjects design.

## Experiment 1

Experiment 1 was conducted to measure the effect of the relative level of temporal sound segments on loudness weights in a condition where only a small, if any, primacy effect was expected. The stimuli were sounds consisting of four 100-ms noise segments, separated by silent gaps of 500-ms duration. The mean level of the first and third segment was varied. It was either identical to the mean level of the remaining segments (baseline condition with a "flat" level profile; [14]), amplified (+5 or +15 dB) relative to the mean level of the remaining segments, or attenuated (−5 or −15 dB). Thus, we compared the effects of attenuation and amplification of a given segment by the same dB value on the temporal weights, and investigated whether this effect differs between the sound onset (level change on segment 1) and later temporal parts of a sound (level change on segment 3).

### Method

**Listeners.** Eight listeners with normal hearing participated in Experiment 1 (8 female, age 20–30 years). They reported no history of hearing problems. Hearing thresholds were measured by Békésy audiometry with pulsed 270-ms pure tones. All listeners showed thresholds

less than or equal to 15 dB HL on both ears in the frequency range between 125 Hz and 8 kHz. All listeners were students from the Johannes Gutenberg-Universität Mainz and received partial course credit for their participation. The experiments were conducted according to the principles expressed in the Declaration of Helsinki. All listeners participated voluntarily after providing informed written consent, after the topic of the study and potential risks had been explained to them. They were uninformed about the experimental hypotheses. The Ethics Committee of the Institute of Psychology of the Johannes Gutenberg-Universität Mainz approved the study (reference number 2016-JGU-psychEK-002).

**Stimuli and apparatus.** Level-fluctuating sounds were presented, each consisting of four 100-ms Gaussian white noise segments. Between each of the segments, there was a silent gap of 500-ms duration. Level fluctuations were created by assigning each segment a sound pressure level independently and at random from a normal distribution on each trial (see section Procedure). In addition to this basic condition where the mean level of all of the four segments was identical (i.e., all segment levels were drawn from the same random distribution), and which is now and in the following termed the *baseline condition*, the mean level of either segment 1 or segment 3 was varied by either attenuating or amplifying it by 5 or 15 dB. When the mean level of segment 1 was changed, no change was applied to the mean level of segment 3, and vice versa. Fig 1 depicts the resulting nine level profiles.

All sounds were generated digitally, D/A-converted by an RME ADI/S with 44.1 kHz sampling frequency and 24 bit resolution, attenuated by a TDT PA5 programmable attenuator, buffered by a TDT HB7 headphone buffer, and presented diotically via Sennheiser HDA 200

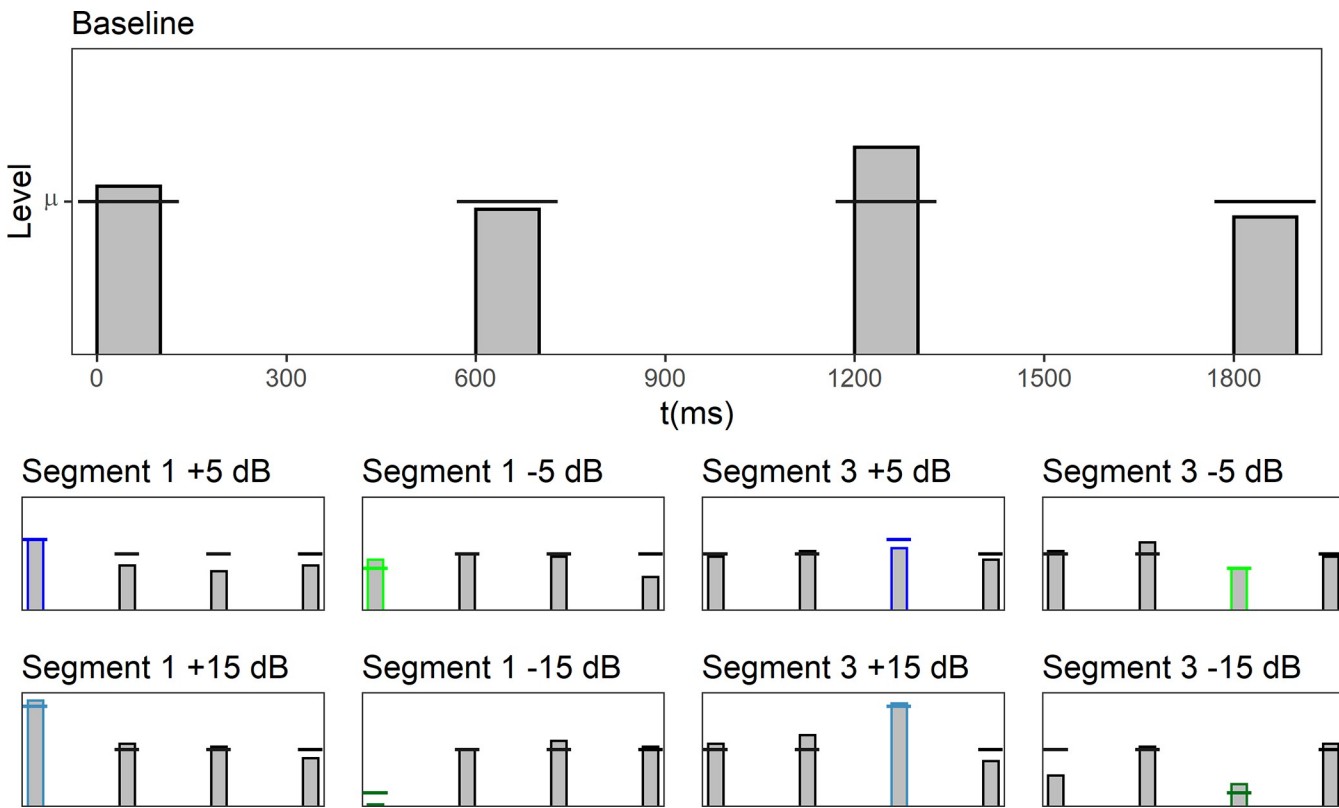

**Fig 1. Level profiles in the nine different conditions presented in Experiment 1.** The stimuli consisted of four 100-ms broadband noise segments. The horizontal lines in each panel display the mean sound pressure of the distribution from which the segment levels were drawn. The bars display the randomly selected segments levels within an example trial.

circumaural headphones. The audio system was calibrated according to [23]. Subjects were tested in a double-walled sound-insulated chamber. Instructions were presented on a computer screen.

**Procedure.** To estimate temporal loudness weights, we used an established experimental paradigm from previous experiments (e.g., [5,8,11,14]). On each trial, a level-fluctuating noise consisting of four segments was presented. The four segment levels were set by drawing each segment's sound pressure level independently and at random from a normal distribution on each trial. On each trial, all segment levels were either sampled from a level distribution with higher mean ($\mu_L$ = 56.125 dB SPL) or a distribution with lower mean ($\mu_S$ = 53.875 dB SPL). The standard deviation was $\sigma$ = 2.5 dB for both distributions. Overly loud or soft segments were avoided by limiting the range of possible sound pressure levels to $\mu \pm 3 \cdot \sigma$.

On each trial, subjects decided whether the presented sound had been louder or softer in comparison to previous trials within the same experimental block. Thus, a one-interval, two-alternative forced-choice (1I, 2AFC) *absolute identification task* [24] with a virtual standard (e.g., [25]) was used. One could also describe it as a *sample discrimination task* [26–28] where the listeners decided whether the segment levels had been drawn from the distribution with a higher mean or from the distribution with a lower mean. Importantly, the listeners were instructed to evaluate the "global" loudness of the entire sound, that is, the loudness across the entire stimulus duration, encompassing all four segments and the silent gaps in between.

The inter-trial interval was 1500 ms, with the restriction that the next trial never started before the response to the preceding trial had been given. Trial-by-trial feedback was given during the first seven trials of each block so that subjects could easily adopt a decision criterion for the new experimental condition. Those trials were not considered for the data analysis. A summarizing feedback was provided each time 50 trials were completed. It contained the number of correct and false answers, percent correct and the number of $\mu_L$ and $\mu_S$ trials as well as the number of "louder" and "softer" responses. Note that a response was classified as correct if the response ("louder"/"softer") matched the mean of the distribution that the stimulus' segment levels were drawn from ($\mu_L/\mu_S$).

We used our usual rule of thumb from previous experiments [9], according to which 100 trials per temporal segment are needed to obtain reliable weight estimates. Thus, we collected 400 trials per condition, resulting in a total of 3600 trials per listener. The required number of trials for each level profile was distributed evenly across five experimental sessions. Within an experimental session, sounds with the same level profile were presented in blocks of 80 trials each. The order of conditions was chosen randomly. In addition to the five experimental sessions, there was an initial session in which hearing levels were measured and practice blocks were presented for all of the nine level profiles. The duration of each session was approximately 60 minutes, including a mandatory pause after 30 minutes.

**Data analysis.** The perceptual weights representing the importance of the temporal segments for the decision in the sample discrimination task were estimated from the trial-by-trial data via multiple logistic regression. This analysis is based on a decision model [14] which assumes that listeners use a decision variable

$$D(\boldsymbol{L}) = \left(\sum_{i=1}^{k} w_i L_i\right) - c, \qquad \text{Eq 1}$$

where $L_i$ is the sound pressure level of segment $i$, $k$ is number of segments, $\boldsymbol{L}$ is the vector of segment levels, $w_i$ is the perceptual weight assigned to segment $i$, and $c$ is a constant representing the decision criterion [cf. 5,29]. The model assumes that a listener responds that the noise presented on a given trial was loud rather than soft if $D(\boldsymbol{L}) > 0$, and that the probability of a

"loud" response is

$$P(''\text{loud}'') = \frac{e^{D(L)}}{1 + e^{D(L)}}.$$

Eq 2

Thus, the decision model assumes that the listener compares a weighted sum of the segment levels to a fixed decision criterion, and responds that the sound was of the "louder" type if the weighted sum exceeds the criterion. If the weighted sum is smaller than the criterion, then the model predicts that the listener classifies the sound as "softer".

In the data analysis, the binary responses ("louder" or "softer") served as the dependent variable. The predictors (i.e., the 4 segment levels) were entered simultaneously. The regression coefficients were taken as the decision weight estimates. For a given segment, a regression coefficient equal to zero means that the segment had no influence at all on the decision. For the same segment, a regression coefficient greater than zero means that the probability of responding that the sound was of the "louder" type increased with the sound pressure level of the segment.

A separate logistic regression model was fitted for each combination of listener and condition (amount and position of attenuation/amplification). A summary measure of the predictive power of a logistic regression model is the area under the Receiver Operating Characteristic (ROC) curve (for details see [30]). Areas of 0.5 and 1.0 correspond to chance performance and perfect performance of the model, respectively. Across the 72 fitted logistic regression models, the area under the ROC curve ranged between 0.61 and 0.91 ($M = 0.78$, $SD = 0.08$), indicating on average reasonably good predictive power [31].

Since the *relative* contributions of the different segments to the decision were of interest rather than the absolute magnitude of the regression coefficients, the four regression coefficients were normalized for each fitted model such that the mean of the absolute values of the three (conditions containing an amplified or attenuated segment) or four (baseline condition) segments that were not attenuated or amplified was 1.0.

To investigate whether attenuation and amplification of a given segment by the same dB value resulted in a change in the segment weight by the same factor, *relative* to the weight assigned to this segment in the baseline condition, we calculated *weight factors*. The regression coefficients were normalized in the same way as outlined above, but slightly negative regression coefficients were set to a value of 0.001. Such slightly negative weights occurred in some cases on the individual level, but were not significantly different from zero. This was done prior to the normalization to ensure that the resulting normalized individual weights were suitable for the calculation of the weight factors. In case of the amplified segments, the weight factor $wf_{i,amp}$ for segment $i$ with amplification *amp* was calculated by dividing the normalized weight on the segment in the condition with amplification ($w_{i,amp}$) by the normalized weight on the corresponding segment in the baseline condition ($w_{i,baseline}$).

$$wf_{i,amp} = \frac{w_{i,amp}}{w_{i,baseline}}$$

Eq 3

In order to facilitate the comparison of the effects of attenuation and amplification on the weight assigned to the segment changed in level, we defined the weight factor for the case of attenuated segments so that a *reduction* in weight corresponded to a weight factor *larger than 1.0*. The weight factor $wf_{i,att}$ for an attenuated segment $i$ with a specified attenuation *att* was calculated by dividing the normalized weight on the segment in the baseline condition ($w_{i,baseline}$) by the normalized weight on the corresponding segment in the condition with attenuation

$(w_{i,att})$.

$$wf_{i,att} = \frac{w_{i,baseline}}{w_{i,att}} \qquad \text{Eq 4}$$

Thus, if for instance an amplification by 10 dB caused an increase in the weight assigned to the amplified segment by a factor of 3 relative to the segment weight in the baseline condition, and an attenuation by 10 dB caused a decrease in weight again by a factor of 3, then with the above definitions of the weight factors, the weight factor would be 3.0 for amplification as well as for attenuation. Thus, using this definition, the effect of amplification or attenuation on the weights is in this case the same for the same dB change (weight change by a factor of 3), although of course in opposite direction (increase when amplified, decrease when attenuated). The weight factors were computed separately for each listener.

The individual normalized temporal weights and the weight factors were analyzed with repeated-measures analyses of variance (rmANOVAs) using a univariate approach with Huynh-Feldt correction for the degrees of freedom [32]. The correction factor $\tilde{\varepsilon}$ is reported, and partial $\eta^2$ is reported as measure of association strength. An α-level of .05 was used for all analyses.

## Results

The average sensitivity in terms of $d'$ is shown in Table 1 for each of the nine conditions. There was a significant effect of condition on $d'$, $F(8, 56) = 2.72$, $\tilde{\varepsilon} = .744$, $p = .026$, $\eta_p^2 = .280$, with the lowest mean sensitivity in the condition where the third segment was amplified by 15 dB, and the highest mean sensitivity in the baseline condition and the condition where the third segment was amplified by 5 dB.

The mean normalized weights are plotted on a logarithmic scale in Fig 2. In the baseline condition (black circles) where none of the segments was attenuated or amplified, the four segment weights descriptively showed no primacy effect, but a small trend for a recency effect. However, an rmANOVA with the within-subjects factor segment number (1–4) on the normalized weights in this baseline condition showed no significant effect of segment number ($p > .14$), indicating that the weights on the four segments did not differ in a systematic way. This was expected, because previous research has shown that the mechanism(s) causing the primacy effect show substantial recovery during silent gaps of more than 350 ms [8]. Therefore, segments that were neither attenuated nor amplified in the present experiment should receive roughly the same weight as the segment at sound onset.

Turning to the conditions where the level of one of the segments was amplified (+5 or +15 dB; blue and blue-grey diamonds), Fig 2 shows that the weights on the corresponding

**Table 1. Mean sensitivity ($d'$) across listeners in the four different conditions of Experiment 1.** $N = 8$.

| Condition | Mean of $d'$ | SD of $d'$ |
|---|---|---|
| Baseline | 0.75 | 0.29 |
| Segment 1 −15 dB | 0.67 | 0.30 |
| Segment 1 −5 dB | 0.74 | 0.27 |
| Segment 1 +15 dB | 0.66 | 0.31 |
| Segment 1 +5 dB | 0.73 | 0.21 |
| Segment 3 −15 dB | 0.66 | 0.25 |
| Segment 3 −5 dB | 0.72 | 0.25 |
| Segment 3 +15 dB | 0.49 | 0.24 |
| Segment 3 +5 dB | 0.75 | 0.33 |

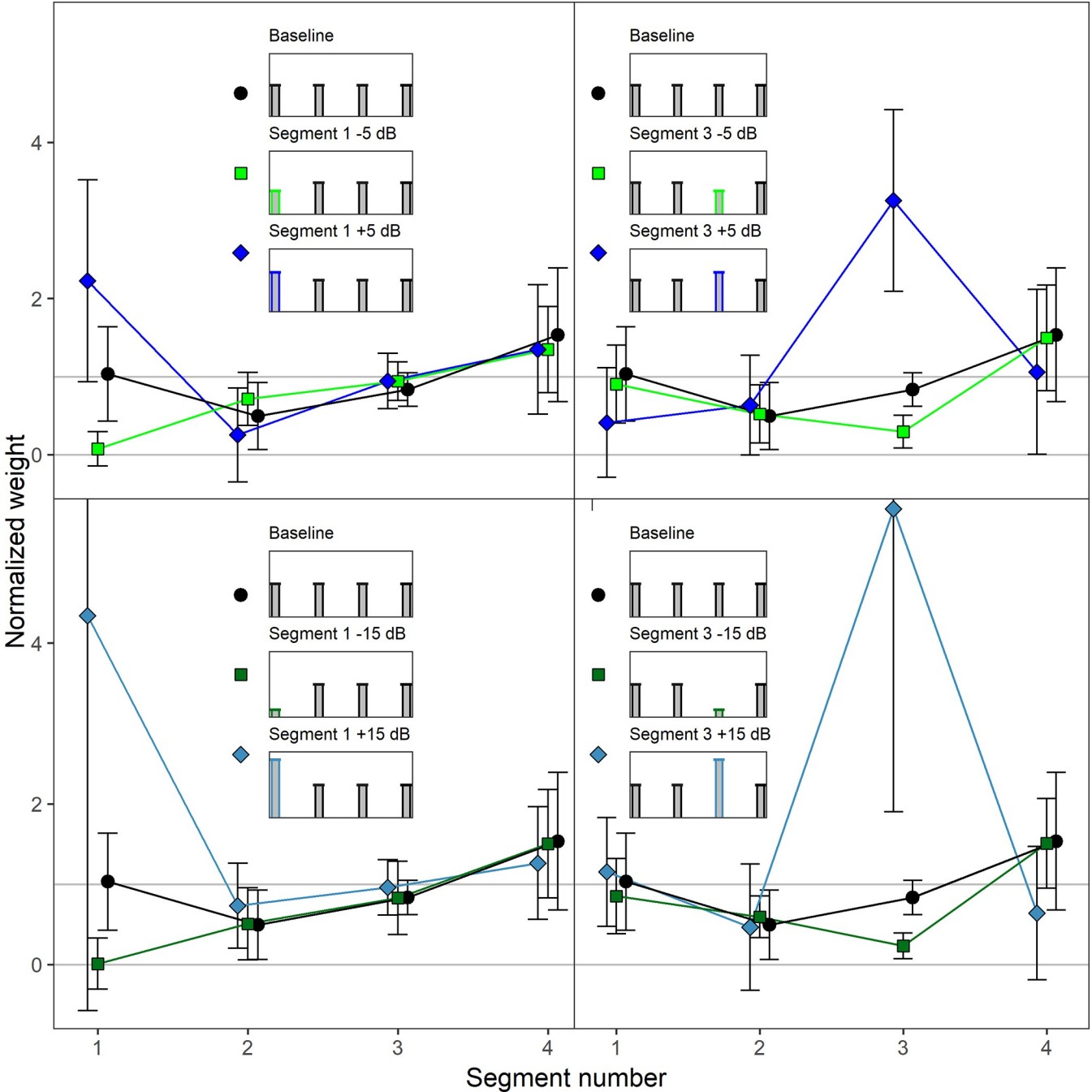

**Fig 2. Experiment 1.** Mean normalized weights as a function of segment number. In each panel, depictions of the mean level profiles of the corresponding stimuli are shown in the upper half. The weights observed in the baseline condition containing no amplified or attenuated segments are plotted in all panels of the figure for reference. Black circles: Baseline condition (mean level identical for all of the four segments). Light green squares: Attenuation by 5 dB. Dark green squares: Attenuation by 15 dB. Blue diamonds: Amplification by 5 dB. Blue-gray diamonds: Amplification by 15 dB. Error bars show 95% confidence intervals (CIs) of the mean. Note that for better visibility, the lines are shifted slightly against each other on the x-axis.

segments increased, relative to the weights of the same segments in the baseline condition. In contrast, if the level of one of the segments was attenuated (−5 or −15 dB; light and dark green squares), the relative weights on that particular segment decreased. This was the expected pattern, compatible with loudness dominance. The pattern of weights on the segments with

unaltered mean level was similar to the pattern in the baseline condition. Thus, despite the temporal gaps between the segments, a loudness dominance effect was observed.

We conducted a repeated-measures ANOVA with the within-subjects factors level profile and segment number (1–4) on the normalized weights. The relevant level profile × segment number interaction was significant, $F(24, 168) = 5.77$, $\tilde{\varepsilon} = .093$, $p = .012$, $\eta_p^2 = .452$, which indicates that the amplification or attenuation of one of the segment levels changed the pattern of weights, as expected.

One of the main aims of Experiment 1 was to answer the question of whether the *relative* change in weights (i.e. the change of an attenuated/amplified segments weight in comparison to the weight of the same segment in the baseline condition) was identical between attenuation and amplification. Furthermore, we wanted to investigate if there was an effect of the position of the level change (that is, a difference between the effects of relative level for a level change on the first versus on the third segment) in the absence of the primacy effect. To answer these questions Table 2 shows the geometric mean of the weight factors for the different positions of the amplified/attenuated segment, directions of the level change, and magnitudes of level change. Note that these weight factors were defined so that both an increase in weight due to an amplification and a decrease in weight due to an attenuation of a given segment corresponded to a weight factor larger than 1.0 (see Eqs 3 and 4 above). For an attenuation or amplification of the third segment, the mean weight factors (i.e., the relative change in weights compared to the baseline condition) were similar in size. In contrast, for an attenuation or amplification of the first segment, attenuation resulted in substantially larger mean weight factors (and standard deviation of the weight factors) than amplification. For an attenuation or amplification of 15 dB, the weight factors were increased compared to an attenuation or amplification of 5 dB.

We conducted an rmANOVA on the log-transformed weight factors, with the within-subjects factors position of level change (segment 1 or 3), direction of level change (attenuation or amplification), and magnitude of level change (5 or 15 dB). There was a significant effect of the direction of level change, $F(1, 7) = 13.16$, $p = .008$, $\eta_p^2 = .653$, $d_z = 1.28$. On average, the weight factor was smaller for amplification compared to attenuation ($M = 4.07$, $SD = 2.91$ and $M = 12.17$, $SD = 5.94$, respectively). The effect of the position of level change was not significant, $F(1, 7) = 4.65$, $p = .068$, $\eta_p^2 = .399$, $d_z = 0.76$. Descriptively, the average weight factor was larger for a level change of the first segment compared to a level change of the third segment ($M = 10.53$, $SD = 6.77$ and $M = 4.70$, $SD = 2.64$, respectively). The effect of the magnitude of level change was not significant, $F(1, 7) = 0.83$, $p = .393$, $\eta_p^2 = .106$, $d_z = 0.32$. Thus, although the mean weight factors were descriptively higher for a level change of 15 dB than for level changes of 5 dB (see Table 2), this effect was not very strong. The direction × position interaction was significant, $F(1, 7) = 22.34$, $p = .002$, $\eta_p^2 = .761$, $d_z = 1.67$. As shown in Table 2, the

**Table 2. Experiment 1.** Geometric means and standard deviations of the weight factors for the different positions, directions, and magnitudes of level change in Experiment 1. $N = 8$.

| Level change | | Weight factor | | | |
| --- | --- | --- | --- | --- | --- |
| *Position (segment)* | *Direction* | *Mean* | | *SD* | |
| | | *5 dB* | *15 dB* | *5 dB* | *15 dB* |
| 1 | amplification | 2.99 | 3.49 | 2.57 | 4.11 |
| 1 | attenuation | 26.17 | 45.17 | 4.07 | 8.87 |
| 3 | amplification | 4.56 | 5.79 | 2.24 | 3.02 |
| 3 | attenuation | 3.62 | 5.11 | 3.24 | 2.4 |

weight factors for segment 1 were higher for attenuation than for amplification, while for segment 3, the weight factors for attenuation were similar to those for amplification. We conducted post-hoc separate paired-samples $t$-tests for attenuation and amplification, each comparing the weight factors of the first and the third segment, averaged across the magnitudes of the level change. For attenuation, the effect of position was significant, $t(7) = 3.98$, $p = .005$, $d_z = 1.41$, two-tailed, whereas for amplification, the effect of position was not significant, $t(7) = 1.19$, $p = .272$, $d_z = 0.42$, two-tailed. The results thus indicate that for amplification, the weight factors are approximately independent of the position of level change, whereas for attenuation this is not the case. Separate post-hoc paired-samples $t$-tests for the first and the third segment, each comparing the weight factors for attenuation and amplification, averaged across the magnitudes of the level change showed a significant effect of direction of level change for the first segment, $t(7) = 5.09$, $p = .001$, $d_z = 1.80$, two-tailed. For the third segment, the effect of direction was not significant, $t(7) = 0.53$, $p = .612$, $d_z = 0.19$, two-tailed. Thus, for the third segment, the effects of amplification and attenuation on the weight factors were approximately symmetric, with attenuation and amplification resulting in similar weight factors, whereas for the first segment, the effect of attenuation was considerably larger than the effect of amplification. All other interactions within the rmANOVA were not significant (all $p > .74$).

Taken together, Experiment 1 showed that even if substantial non-simultaneous masking effects on the intensity resolution were not likely to be present (due to the 500-ms gaps between the noise bursts), effects of relative level (dominance) do occur. Furthermore, in the absence of a primacy effect, we observed an asymmetry between amplification and attenuation which was also interacting with a position effect. Attenuation of the first segment led to a larger change in the weight factor (weight relative to the weight in the baseline condition) than amplification of the first segment or attenuation of the third segment did.

## Experiment 2

In Experiment 2, we presented sounds consisting of contiguous segments, for which–in contrast to Experiment 1—a pronounced primacy effect was expected, based on previous studies. This was done to i) assess the potential interactions between loudness dominance and primacy effects and ii) to examine the influence of effects of non-simultaneous masking on the effects of relative level. Again, the level of a single segment within the whole sound was either attenuated or amplified and the position of the level change was varied between two positions between conditions to assess potential interactions.

In contrast to Experiment 1, where no primacy effect was present, in Experiment 2 we expected that the effect of relative level might interact with the primacy effect. A descriptive comparison of the results of the two experiments is performed to answer the question whether the primacy effect and loudness dominance do interact. In addition, non-simultaneous-masking effects on the intensity resolution were expected to occur in Experiment 2 because the segments were presented without silent gaps between the segments. By comparing the loudness dominance effects of Experiment 1 and Experiment 2, we were thus able to gain insight into whether these masking effects play a role in the loudness dominance effect.

### Method

**Listeners.**   In this experiment, 10 listeners with normal hearing participated (8 female, 2 male, age 20–28 years). None of them had participated in Experiment 1. The same standards for the recruitment and participation of the listeners and for the conduction of the experiments were applied as in Experiment 1.

**Stimuli.** Level-fluctuating sounds were presented, either consisting of four or ten 100-ms Gaussian white noise segments. Level fluctuations were created by drawing each segment's sound pressure level independently and at random from a normal distribution each trial (see section Procedure).

In addition to these two baseline conditions where the mean level of all of the four or ten segments was identical (i.e., all segment levels were drawn from the same random distribution), the mean level of either segment 1 or segment 3 (four-segments sounds), or segment 1 or segment 7 (ten-segments sounds), was varied by either attenuating or amplifying it by 15 dB. When the mean level of the first segment was changed, no change was applied to segment 3 or 7, and vice versa. Fig 3 depicts the resulting ten level profiles.

**Apparatus, procedure and data analysis.** Apparatus, procedure and data analysis were largely the same as in Experiment 1 and thus only deviations are reported here.

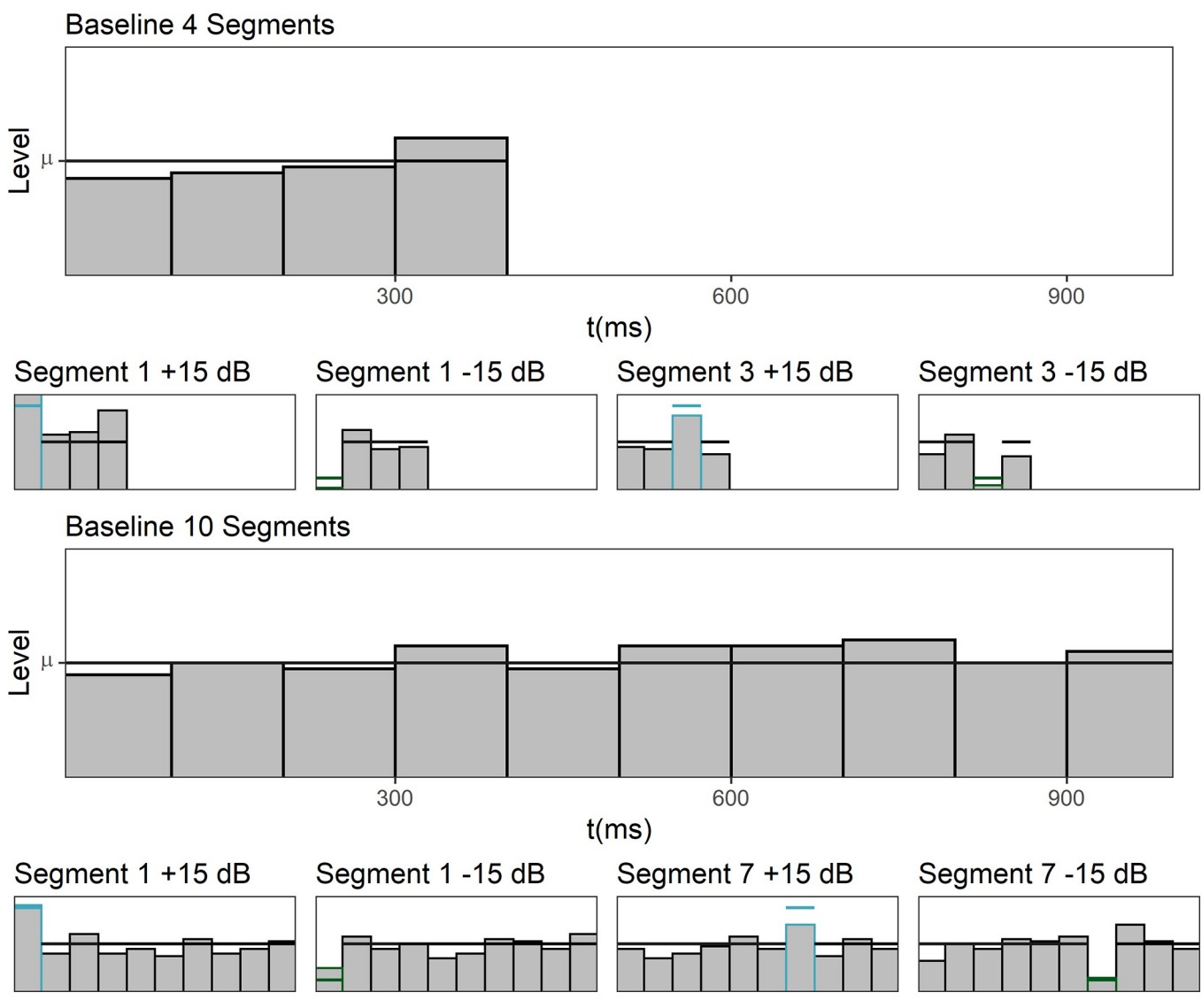

**Fig 3. Experiment 2.** Level profiles in the ten different conditions. The horizontal lines in each panel display the mean sound pressure level of the distribution from which the segment levels were drawn. The bars display the randomly selected segment levels within an example trial.

Data collection this time required 400 or 1000 trials per condition, depending on the number of segments of the sounds in the corresponding condition. This resulted in a total of 7000 trials per listener which were distributed evenly across six experimental sessions.

Again, a separate logistic regression model was fitted for each combination of listener and condition. Across the 100 fitted logistic regression models, the area under the ROC curve ranged between 0.62 and 0.93 ($M = 0.85$, $SD = 0.07$), and thus was comparable to the values from Experiment 1.

## Results

The average sensitivity in terms of $d'$ is shown in Table 3 for each of the nine conditions. There was a significant effect of condition on $d'$, $F(9, 81) = 15.04$, $\tilde{\varepsilon} = .558$, $p < .001$, $\eta_p^2 = .626$, with the lowest mean sensitivity in the conditions with four segments where the third or the first segment were amplified by 15 dB, and the highest mean sensitivity in the ten-segment baseline condition.

Fig 4 shows the mean normalized weights in the ten conditions. Note that due to the higher maximum weight measured in this experiment, the weight range shown in Fig 4 is larger than in Fig 2. Like in Experiment 1, the weights were normalized so that the mean of the absolute values of the weights on the unattenuated/unamplified segments within each condition for each listener was 1.0. In the baseline condition (black circles), where each segment had the same mean level, there was a primacy effect in the sense that the first segment received the highest weight and the weights on the following segment decreased as a function of segment number. This occured both for the four- and the ten-segment-sounds. We conducted separate rmANOVAs for the four- and the ten-segment sounds, with the within-subjects factor segment number (1–4 or 1–10, respectively) on the normalized weights in the baseline condition. For both the four-segment sounds, $F(3, 27) = 8.00$, $\tilde{\varepsilon} = .819$, $p = .001$, $\eta_p^2 = .471$, and the ten-segment sounds, $F(9, 81) = 12.18$, $\tilde{\varepsilon} = .482$, $p < .001$, $\eta_p^2 = .575$, the main effect of segment number was significant, indicating that the weights differed between the segments.

In the conditions with attenuated or amplified segments (blue diamonds or green squares), Fig 4 shows that for both the four- and the ten-segment sounds, and for both the first and the third or the seventh segment, amplification resulted in a substantial increase in the weight on the amplified segment and a small reduction of the weights on the neighboring segments, relative to the other segments with unaltered level. For attenuation, the weights on the attenuated segments decreased and in the case of an attenuation of the first segment, the weights on the following segments slightly increased, showing a "delayed" primacy effect [33].

**Table 3. Mean sensitivity ($d'$) across listeners in the ten different conditions of Experiment 2.** $N = 10$.

| Condition | Mean of $d'$ | SD of $d'$ |
|---|---|---|
| **10 Segments baseline** | 1.21 | 0.33 |
| **10 Segments Segment 1 −15 dB** | 1.15 | 0.30 |
| **10 Segments Segment 1 +15 dB** | 0.85 | 0.21 |
| **10 Segments Segment 7 −15 dB** | 1.20 | 0.37 |
| **10 Segments Segment 7 +15 dB** | 0.81 | 0.19 |
| **4 Segments baseline** | 0.96 | 0.27 |
| **4 Segments Segment 1 −15 dB** | 0.90 | 0.22 |
| **4 Segments Segment 1 +15 dB** | 0.78 | 0.19 |
| **4 Segments Segment 3 −15 dB** | 1.00 | 0.17 |
| **4 Segments Segment 3 +15 dB** | 0.78 | 0.16 |

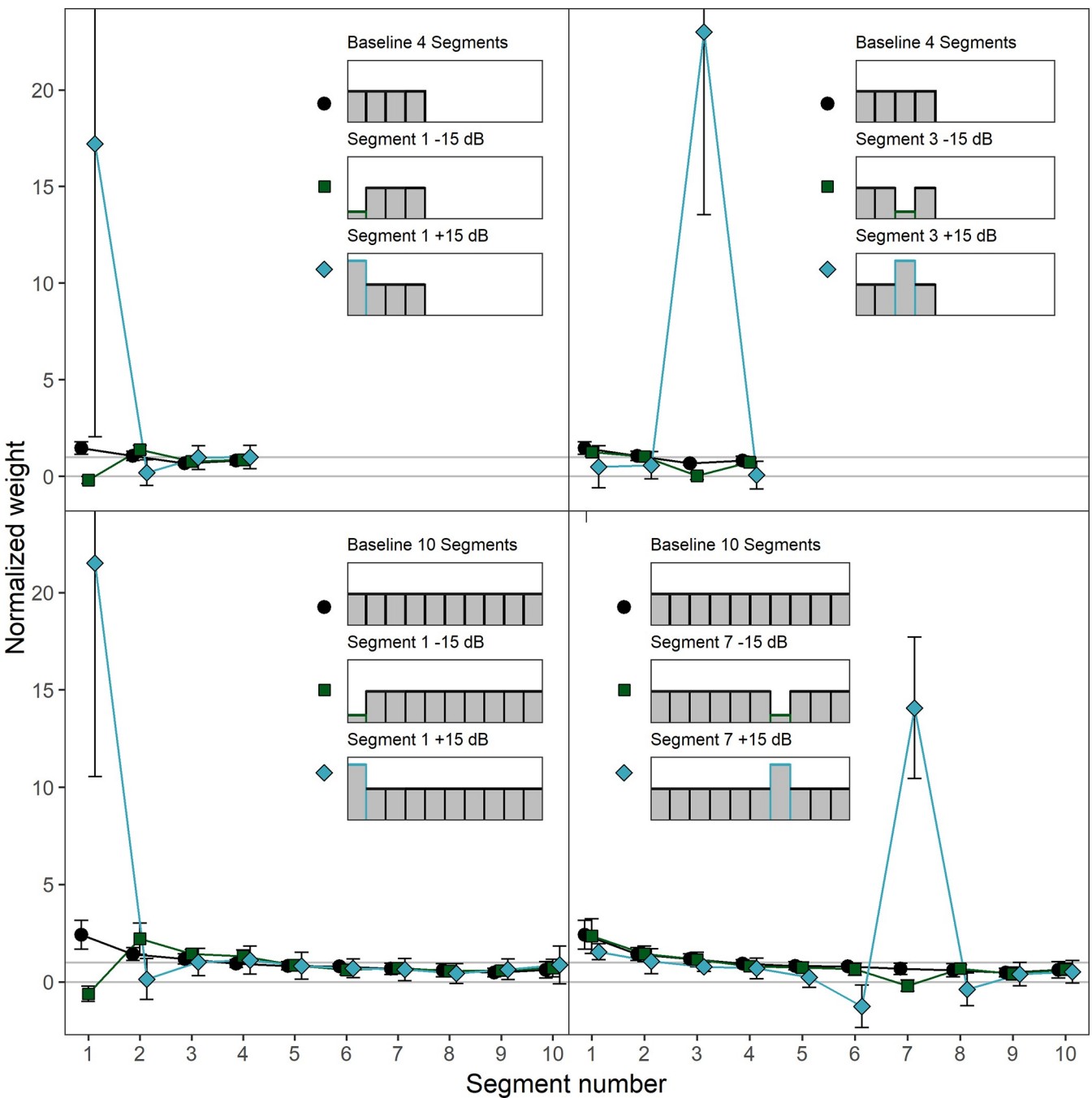

**Fig 4. Experiment 2.** Mean normalized weights as a function of segment number for the four- and ten-segment sounds. Upper panels: Four-segment sounds. Lower panels: Ten-segment sounds. Left panels: Attenuation/amplification of segment 1. Right panels: Attenuation/amplification of segment 3 (four-segment sounds) or 7 (ten-segment sounds). In each panel, depictions of the mean level profiles of the corresponding stimuli are shown in the upper half. The weights observed in the respective baseline condition in which none of the segments were attenuated or amplified are plotted in all panels of the figure for reference. Black circles: Baseline. Dark green squares: Attenuation by 15 dB. Blue-gray diamonds: Amplification by 15 dB. Error bars show 95% confidence intervals (CIs). Note that for better visibility, the lines are shifted slightly against each other on the x-axis.

We conducted two separate rmANOVAs on the normalized weights on the four- and the ten-segment sounds. The within-subjects factors were level profile (5 level profiles for each rmANOVA) and segment number (1–4 in the rmANOVA for the four-segment sounds, 1–10

**Table 4. Experiment 2.** Geometric means and standard deviations of the weight factors for the different positions and directions of level change, separately for the four- and ten-segment sounds.

| Level change | | Weight factor | | | |
| Position | Direction | Mean | | SD | |
| | | 4 Segments | 10 Segments | 4 Segments | 10 Segments |
| 1 | amplification | 9.91 | 8.82 | 2.32 | 1.74 |
| 1 | attenuation | 309.93 | 245.26 | 3.25 | 2.16 |
| 3/7 | amplification | 45.54 | 39.04 | 2.89 | 3.56 |
| 3/7 | attenuation | 29.83 | 29.93 | 6.6 | 5.98 |

in the rmANOVA for the ten-segment sounds). The level profile × segment number interaction was significant for both the four-segment sounds, $F(12, 108) = 13.68$, $\tilde{\varepsilon} = .171$, $p < .001$, $\eta_p^2 = .603$, and the ten-segment sounds, $F(36, 324) = 22.81$, $\tilde{\varepsilon} = .039$, $p < .001$, $\eta_p^2 = .717$. This indicates that the pattern of segment weights differed between the different level profiles, as expected.

To investigate whether the relative change in weights on the attenuated or amplified segment was comparable between attenuation and amplification as well as for the first and for the middle third or seventh segment, the geometric means of the weight factors are displayed in Table 4. We conducted an rmANOVA on the log-transformed weight factors, with the within-subjects factors direction of level change (attenuation, amplification), position of level change (first segment, middle segment), and number of segments (4 or 10). There was a significant effect of the direction of level change, $F(1, 9) = 37.25$, $p < .001$, $\eta_p^2 = .805$, $d_z = 1.93$. Attenuation resulted in a higher mean weight factor ($M = 90.76$, $SD = 6.14$) than amplification ($M = 19.85$, $SD = 3.34$), as in Experiment 1. The direction × position interaction was also significant, $F(1, 9) = 43.90$, $p < .001$, $\eta_p^2 = .830$, $d_z = 2.10$. As shown in Table 4, attenuation at the beginning of the sound resulted in higher weight factors compared to attenuation the middle position. For amplification, the pattern was reversed, with amplification at the beginning of the sound resulting in smaller weight factors compared to amplification at the middle position. We conducted separate paired-samples $t$-tests for attenuation and amplification, each comparing the weight factors of the first segment and the middle segment (segment 3 and 7), averaged across the different number of segments (4 and 10). The effect of position on the weight factors was significant for the amplified segments, $t(9) = 4.38$, $p = .002$, $d_z = 1.39$, as well as for the attenuated segments, $t(9) = 4.77$, $p = .001$, $d_z = 1.51$. Thus, for attenuation as well as for amplification, there was a significant position effect on the weight factors. However, the direction of the effect differed between attenuation and amplification, as pointed out above.

Taken together, the results from Experiment 2 indicate that the effects of relative level for contiguous sounds were neither symmetrical for attenuation and amplification, nor for different temporal positions of the amplified/attenuated segment. On average, attenuation of a segment resulted in a stronger change of the weight on the segment in comparison to the baseline condition than amplification did. In addition, the effects of an attenuation were more pronounced when the attenuated segment was at the beginning of the sound, compared to an attenuation of a middle segment. Both these findings are in line with the results from Experiment 1. In contrast, amplification resulted in a larger change in the weight factor for the amplified segment when the mean level of a middle segment was changed, compared to the effect of an amplification of the first segment. This is a difference to Experiment 1, where no such position effect of amplification was observed. The results thus indicate an interaction of the primacy effect and loudness dominance for amplification, but not for attenuation. We will return to this point in the discussion.

In comparison to Experiment 1, in which we presented segments separated by 500 ms silent gaps, the effects of relative level were more pronounced in Experiment 2, as indicated by the weight factors shown in Tables 2 and 4. An unpaired t-test comparing the average weight factors averaged across both positions and across both attenuation and amplification for each listener in Experiment 1 to the averaged weight factors of the listeners in Experiment 2 showed a significant difference between the average weight factors of the two experiments, $t(15.6) = 5.08$, $p < .001$, $d_z = 1.28$ (Experiment 1: $M = 8.26$, $SD = 1.97$; Experiment 2: $M = 45.19$, $SD = 2.08$). Because except for the gaps between the segments, all of the parameters of the sounds were identical between the two experiments, this suggests that the effects of relative level differ between contiguous sounds and sounds encompassing silent gaps. A potential explanation of this finding is that effects of non-simultaneous masking on intensity resolution play a role in loudness dominance, as we discuss in greater detail below.

## Experiment 3

We conducted Experiment 3 to further investigate the effects of inter-segment gaps on the loudness dominance effect (i.e., the effect of relative level on the segment weights). In Experiment 1, where the segments were separated by 500-ms silent gaps, the weight factors were on average significantly lower compared to the weight factors in Experiment 2, where contiguous sounds were presented. However, two different groups of listeners participated in the two experiments, and perceptual weights often show considerable inter-individual differences [34]. In Experiment 3, we therefore investigated the effect of inter-segment gaps on the weight factors in a within-subjects design. Each listener evaluated the global loudness of sounds consisting of four contiguous segments, as well as of sounds consisting of four segments that were separated by 500 ms gaps. This provided a stronger test of the hypothesis that inter-segment gaps reduce the dominance effect. In addition, we also presented contiguous sounds that had the same total duration as the sounds including silent gaps, to investigate whether potential differences between the former two sound types may be due to effects of total sound duration. In Experiment 2, the total sound duration of the four-segment sounds without gaps was shorter than for the four-segment sounds with gaps presented in Experiment 1, so that the sound duration and the presence/absence of inter-segment gaps had not been varied independently.

### Method

**Listeners.**   In Experiment 3, nine listeners with normal hearing participated (7 female, 2 male, age 20–47 years). None of them had participated in Experiment 1 or Experiment 2. The same standards for the recruitment and participation of the listeners and for the conduction of the experiments were applied as in Experiment 1 and Experiment 2.

**Stimuli.**   Three different types of level-fluctuating sounds were presented, either consisting of four 100-ms or 475-ms contiguous Gaussian white noise segments, or of four 100-ms Gaussian white noise segments separated by 500-ms gaps. For each of these types of stimuli, a "baseline"-condition in which all the segments had the same mean level was presented. In addition, we presented a level profile where the third segment was attenuated by 15 dB, and a level profile where the third segment was amplified by 15 dB. Fig 5 shows schematic depictions of the resulting nine conditions. The stimulus generation, including the random level fluctuations, was identical with the procedure used in Experiment 1 and 2.

**Apparatus, procedure and data analysis.**   Apparatus, procedure and data analysis were identical to the corresponding conditions in Experiment 1 and 2. Data collection this time required 400 trials per condition. This resulted in a total of 3600 trials per listener which were

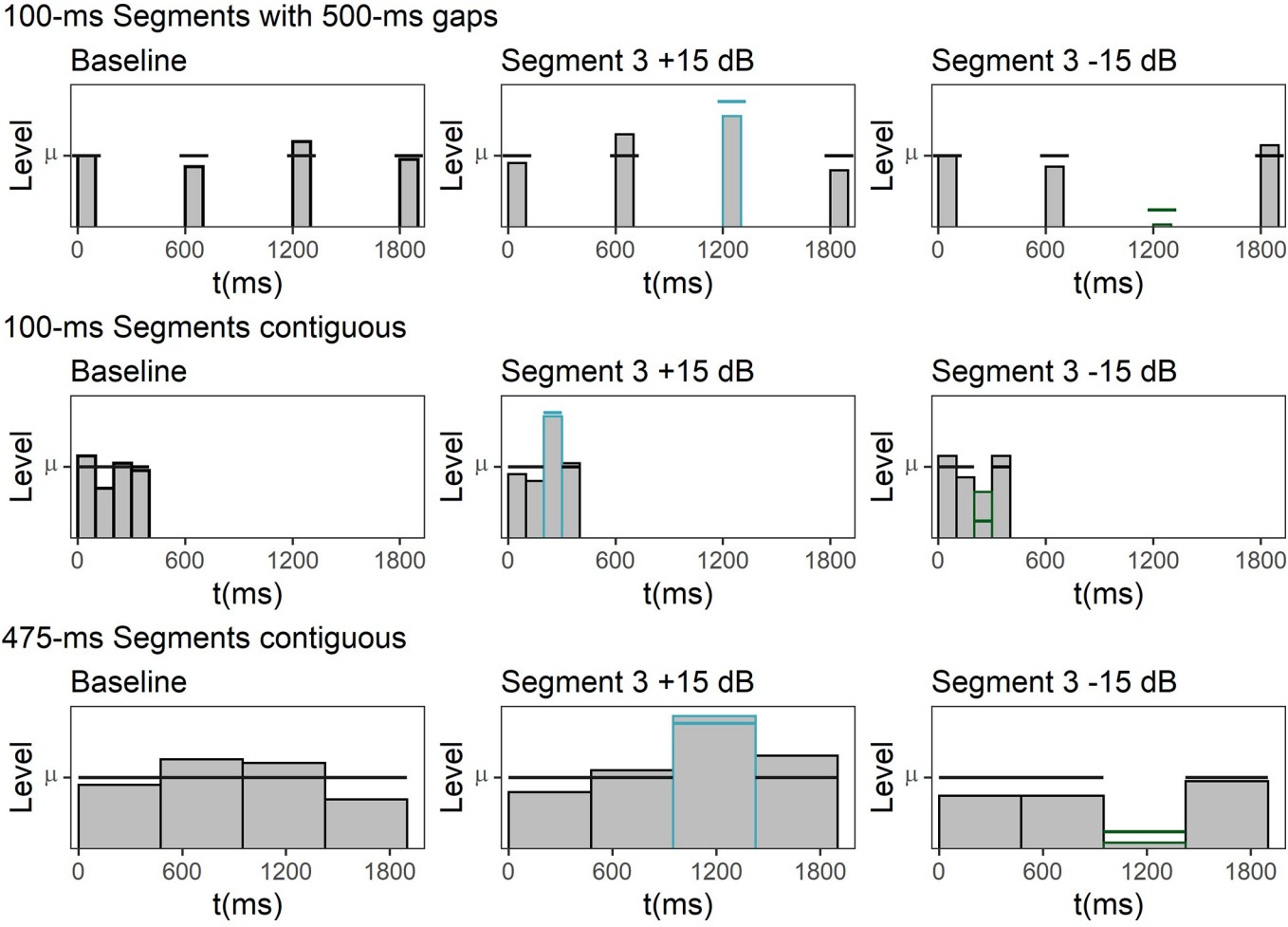

**Fig 5. Experiment 3.** Level profiles in the nine different conditions. The horizontal lines in each panel display the mean sound pressure level of the distribution from which the segment levels were drawn. The bars display the randomly selected segments levels within an example trial.

distributed evenly across four experimental sessions. Again, a separate logistic regression model was fitted for each combination of listener and pause duration. Across the 81 fitted logistic regression models, the area under the ROC curve ranged between 0.68 and 0.92 ($M = 0.83$, $SD = 0.05$), and thus was comparable to the values from Experiment 1 and Experiment 2.

## Results

The average sensitivity in terms of $d'$ is shown in Table 5 for each of the nine conditions. There was a significant effect of condition on $d'$, $F(8, 64) = 6.91$, $\tilde{\varepsilon} = 1$, $p < .001$, $\eta_p^2 = .463$, with the lowest mean sensitivity in the condition with 100-ms segments with 500-ms gaps with an amplification of 15 dB of the level of the third segment and, the highest mean sensitivity in the baseline condition of the same sound type (100-ms Segments with 500-ms gaps).

Fig 6 shows the mean normalized weights in the nine conditions. Note that due to the measured maximum weight in this Experiment, the weight range shown in Fig 6 is larger than in Fig 2 but smaller than in Fig 4. The weights were normalized in the same manner as in Experiment 1 and Experiment 2 so that the mean of the absolute values of the weights on the

**Table 5. Mean sensitivity across listeners (*d′*) in the nine different conditions of Experiment 3.** *N* = 9.

| Condition | Mean of d′ | SD of d′ |
|---|---|---|
| **100-ms Segments contiguous Baseline** | 0.98 | 0.23 |
| **100-ms Segments contiguous Segment 3 −15 dB** | 0.98 | 0.12 |
| **100-ms Segments contiguous Segment 3 +15 dB** | 0.74 | 0.16 |
| **475-ms Segments contiguous Baseline** | 0.95 | 0.20 |
| **475-ms Segments contiguous Segment 3 −15 dB** | 0.92 | 0.11 |
| **475-ms Segments contiguous Segment 3 +15 dB** | 0.72 | 0.14 |
| **100-ms Segments 500 ms-gaps Baseline** | 1.00 | 0.17 |
| **100-ms Segments 500 ms-gaps Segment 3 −15 dB** | 0.88 | 0.28 |
| **100-ms Segments 500 ms-gaps Segment 3 +15 dB** | 0.68 | 0.14 |

unattenuated/unamplified segments within each condition for each listener was 1.0. For the sounds consisting of 100-ms segments separated by 500-ms gaps (left panel of Fig 6) the weights in the baseline condition where none of the segments were attenuated or amplified showed no primacy effect, but a small trend for a recency effect. In contrast, the weights in the baseline conditions of the sounds with 100-ms contiguous segments (middle panel of Fig 6) showed a clear primacy effect with highest weight at sound onset and the weights on the following segments declining over time. For the contiguous sounds composed of 475-ms segments (right panel of Fig 6), the first segment received the highest weight, but the weights assigned to the following segments did not show sizeable differences. This is in accordance with previous studies investigating temporal weighting patterns for comparably long segment durations [9]. In the baseline conditions, separate rmANOVAs for each sound type on the

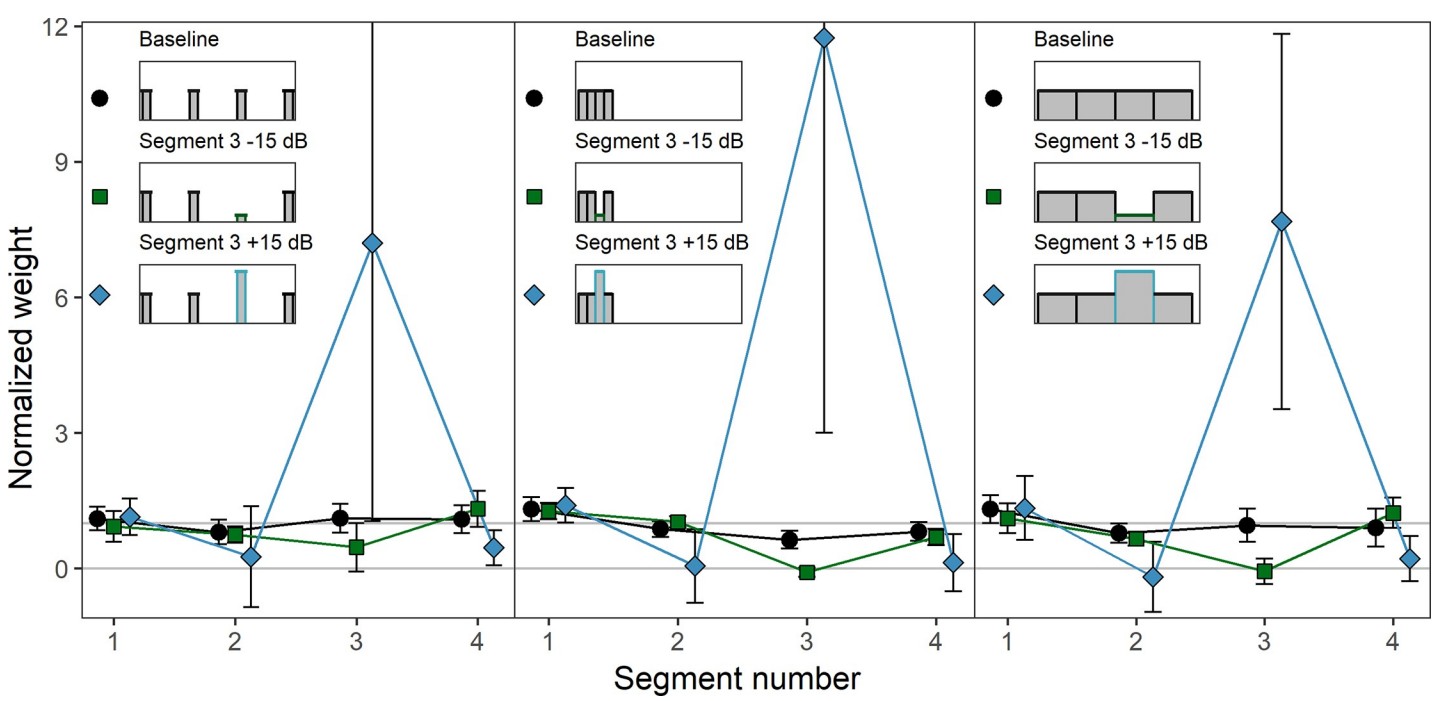

**Fig 6. Experiment 3.** Mean normalized weights as a function of segment number for the three sound types. Left panel: 100-ms segments with 500-ms gaps. Middle panel: 100-ms contiguous segments. Right panel: 475-ms contiguous segments. Black circles: Baseline condition. Dark green squares: Attenuation of segment 3 by 15 dB. Blue-gray diamonds: Amplification of segment 3 by 15 dB. Error bars show 95% confidence intervals (CIs).

segment weights, with the within-subjects factor segment number (1–4), showed neither a significant effect of segment number for the sounds with 500-ms gaps, nor for the contiguous sounds with 475-ms segment duration ($p$ = .36 and $p$ = .17, respectively). In contrast, for the 100-ms contiguous segment sounds, the effect of segment number was significant, $F(3, 24)$ = 7.55, $\tilde{\varepsilon}$ = .867, $p$ = .002, $\eta_p^2$ = .486. This indicates that the temporal weights on the four segments in the baseline condition (flat level profile) only differed in a systematic way for the contiguous sounds with 100-ms segment duration. This is the expected result because the primacy effect has been shown to recover during silent gaps [8], and because the primacy effect has been shown to be reduced for sounds with longer segment durations, which can be explained by assuming that the weight assigned to a given the segments is the integral of an exponential decay function across the segment duration [9].

Again, as in Experiment 1 and Experiment 2, the weights on the amplified segments were increased relative to the baseline condition, while the weights on the attenuated segments were decreased. We conducted an rmANOVA on the normalized weights. The within-subjects factors were sound type (100-ms contiguous segments, 475-ms contiguous segments, 100-ms segments separated by 500-ms gaps), level profile (three level profiles for each rmANOVA) and segment number (1–4). The level profile × segment number interaction was significant, $F(6, 48)$ = 16.75, $\tilde{\varepsilon}$ = .179, $p$ = .003, $\eta_p^2$ = .677, which indicates that the pattern of the weights differed between the different level profiles. The differences between the weighting patterns observed with the different level profiles were not strongly affected by the type of sound; the level profile × segment number × sound type interaction was non-significant, $F(12, 96)$ = 1.18, $\tilde{\varepsilon}$ = .156, $p$ = .332, $\eta_p^2$ = .128. This does not imply that there was no effect of the sound type on the amount by which the weight on an amplified or attenuated segment changed as the dependent variable in the analysis of variance were the normalized weights and therefore only absolute changes within the weights but not the relative changes with respect to the baseline were analyzed. Especially for the attenuated segments, small absolute changes in the weights between sound types can result in substantial changes if those changes are put in relation to the baseline.

Therefore, to answer the primary question of Experiment 3, whether the relative change in weights on the attenuated or amplified segment was more pronounced for contiguous sounds compared to sounds with gaps between the segments, the geometric means of the weight factors are displayed in Table 6. For both attenuation and amplification, the weight factors for the three sound types showed some variation, with the lowest weight factors resulting for both attenuation and amplification for the sounds with 100-ms segments separated by gaps. The variation of the weight factors between the sound types was more pronounced for attenuation than for amplification. For the sounds with 100-ms segments separated by gaps, the weight factors for attenuation and amplification were roughly comparable whereas for the sounds with contiguous 100-ms and 475-ms segments the weight factors under attenuation were a multiple of the weight factors for amplification.

**Table 6. Experiment 3.** Geometric means and standard deviations of the weight factors for the three different sound types separately for the two different directions of level change.

| Sound type | | Weight factor | | | |
|---|---|---|---|---|---|
| Segment Duration (ms) | Gap duration (ms) | Mean | | SD | |
| | | Attenuation | Amplification | Attenuation | Amplification |
| 100 | 0 | 107.3 | 16.12 | 4.53 | 2.32 |
| 100 | 500 | 5.52 | 4.5 | 5.66 | 9.89 |
| 475 | 0 | 157.88 | 9.35 | 6.97 | 3.51 |

We conducted an rmANOVA on the log-transformed weight factors, with the within-subjects factors direction of level change (attenuation, amplification), and sound type (100-ms segments with 500-ms gaps, contiguous 100-ms segments, contiguous 475-ms segments). There was a significant effect of the sound type, $F(2, 16) = 9.76$, $\tilde{\varepsilon} = 1$, $p = .002$, $\eta_p^2 = .550$. The 100-ms segments sounds with 500-ms gaps resulted in the smallest mean weight factor ($M = 4.98$, $SD = 7.20$), while the mean weight factors for the contiguous sounds were higher and quite similar (100-ms segments: $M = 41.59$, $SD = 4.64$; 475-ms segments: $M = 38.42$, $SD = 8.60$). There was also a significant effect of the direction of level change, $F(1, 8) = 23.37$, $p = .001$, $\eta_p^2 = .745$, $d_z = 1.61$. Attenuation resulted in a higher mean weight factor ($M = 45.39$, $SD = 9.60$) than amplification ($M = 8.79$, $SD = 5.02$). The sound type × direction of level change interaction was also significant, $F(2, 16) = 9.61$, $\tilde{\varepsilon} = .859$, $p = .003$, $\eta_p^2 = .546$, which confirms the descriptive differences in the effects of amplification and attenuation on the weight factors for the three sound types. To investigate whether the effect of the direction of level change was statistically significant for each sound type, we conducted separate post-hoc paired-samples $t$-tests for the three sound types, each comparing the weight factors for attenuation and amplification. For both the sounds with contiguous 100-ms and 475-ms segments, the effect of direction of level change on the weight factors was significant, $t(8) = 3.45$, $p = .009$, $d_z = 1.15$, two-tailed; $t(8) = 5.78$, $p < .001$, $d_z = 1.93$, two-tailed, whereas for the sounds composed of 100-ms segments separated by 500-ms gaps, the effect of direction on the weight factors was not significant, $t(8) = 0.49$, $p = .64$, $d_z = 0.16$, two-tailed. As can be seen in Table 6, for both types of contiguous sounds, an attenuation of segment 3 resulted in higher weight factors compared to an amplification. To investigate whether the three sound types differed in their weight factors for both an amplification and an attenuation, we conducted separate post-hoc paired-samples $t$-tests for amplification and attenuation, comparing each of the sound types with each other. In the conditions with an attenuated segment, the weight factors were significantly higher for both the 100-ms contiguous segments, $t(8) = 4.21$, $p = .003$, $d_z = 1.40$, two-tailed, and the 475-ms segments sounds, $t(8) = 5.43$, $p < .001$, $d_z = 1.81$, two-tailed, compared to the sound type with silent gaps between the segments. For amplification, sounds with contiguous 100-ms segments showed a trend towards higher weight factors compared to sounds with 100-ms segments separated by gaps, $t(8) = 1.96$, $p = .086$, $d_z = 0.65$, two-tailed and sounds with contiguous 475-ms segments, $t(8) = 2.16$, $p = .063$, $d_z = 0.72$ two-tailed.

The results confirm the assumption that the observed differences between the weight factors in Experiment 1 and Experiment 2 were in fact due to the gaps between the noise bursts presented in Experiment 1. However, this effect on the weight factors was statistically significant only for an attenuation of one of the sound segments, whereas for amplification, the presence or absence of gaps within a sound did not affect the weight factors in a statistically significant way.

## General discussion

In three experiments, the effects of attenuating or amplifying single temporal segments within a longer sound on temporal loudness weights were examined. In all three experiments and in accordance with previous research [3,11–15], amplified segments with a higher mean level than the remaining segments received an increased weight. Attenuation resulted in the opposite pattern, i.e., in a decreased weight on the attenuated segment. Thus, the present study showed the expected loudness dominance effect.

Averaged across all conditions and experiments, the factor by which the weight on an amplified segment increased relative to the baseline condition was about a sixth of the factor by which the weight on an attenuated segment decreased relative to the baseline condition.

Thus, the average effects of amplification and attenuation on the weight factors were asymmetric. For Experiment 1 and Experiment 2, this was mainly due to the large decrease in the segment weight when the first segment was attenuated. For the middle segment in Experiment 1 (sounds with gaps), the effects of attenuation and amplification on the weight factors were quite symmetrical, whereas for the middle segment in Experiment 2 (contiguous sounds), the effect of amplification on the weight factors was larger than the effect of attenuation. However, in Experiment 3, the effect of an attenuation of the middle segment for contiguous sounds exceeded that of an amplification, showing some deviations from the results of Experiment 2. Thus, although the general effects of relative level were consistent across experiments (in all three experiments, attenuation resulted in decreased weights and amplification resulted in increased weights on the amplified segments), the relative size of the effects of attenuation and amplification showed some variation. Because the sounds with four contiguous 100-ms segments were identical in Experiment 2 and Experiment 3 and yet the size of the effects of amplification and attenuation on the weight factors differed between the experiments, this variation might be due to the different sets of listeners within the experiments, compatible with the previous reports that inter-individual differences in perceptual weights can be quite substantial (see also [9,14,34]).

The effect of the position of level change, i.e. that a change in mean level of the first or a middle segment may or may not result in a comparable change of the weight for the segment, relative to the weight for the corresponding segment in the baseline, was assessed in Experiments 1 and 2 only. In Experiment 1, the effect of a level change on the first segment on the weight factors was similar to the effect of a level change on the third segment for amplification. For attenuation, the effect on the weight factors was stronger when the level change was imposed on the first compared to the third segment. In Experiment 2, there was an effect of the position of level change on the weight factors for amplification as well as for attenuation. The position effect for amplification mainly resulted from the difference between the weight on the first segment in the baseline conditions, which due to the primacy effect was higher compared to the weight on the middle segment in the baseline conditions. At the same time, the weights on the amplified segments did not differ substantially between the first and the middle temporal position. For attenuation, the effect of the position of the level change resulted from lower weights on the attenuated segments when the first segment was attenuated, compared to an attenuation of a middle segment, which was also observed in Experiment 1 and the increase of the weights on the first segment in the baseline condition due to the primacy effect as mentioned above.

Remarkably, the magnitude of the level change, i.e. the amount of attenuation or amplification, did not affect the overall size of the weight factors in Experiment 1 in a statistically significant way. A change in relative segment level of plus or minus 5 dB did not result in a substantially smaller dominance effect than a level change of plus or minus 15 dB, although descriptively, the effect of amplification or attenuation of a single segment on the weight factors was more pronounced for the higher magnitude of level change. Lutfi and Jesteadt [13] reported the effect of relative level in sequences of tones separated by 200-ms gaps to increase when the level differences between the louder and softer tones increased from 1 to 10 dB but found no substantial further increase in the effect at a level difference of 40 dB. Oberfeld, Kuta [15] showed that for a sequence of tones separated by 100-ms gaps, the loudness dominance effect was more pronounced for an attenuation/amplification of 25 dB compared to 5 dB but did not further increase when the difference in mean level between the softer and louder temporal components was 55 dB. For ten-segment sounds consisting of contiguous 100-ms segments, Oberfeld [35] compared the temporal loudness weights on a flat level profile to the weights on level profiles with a 3-, 6-, or 9-dB increase in level across the first three segments.

Except for the maximum attenuation of 3 dB, the weights differed significantly from the weights in the baseline condition. Taken together, the present study and previous experiments suggest that the effect of relative segment level saturates at an amplification or attenuation of between 10 and 20 dB. It would be interesting to further map out the dependence of the loudness dominance effect on the magnitude of the level changes in the range between 10 and 25 dB in future studies.

Also, the effects of relative level were present for sounds that were contiguous as well as for sounds that contained silent gaps between the temporal segments. However, both a between-subjects comparison (Experiment 1 versus Experiment 2) and a within-subjects comparison (Experiment 3) showed that the effects of relative level on the weight factors were larger for the contiguous sounds. This finding might be attributed to forward and backward masking effects, which will be discussed in the Subsection "Loudness dominance, primacy effect and masking effects" below. The effect of relative level did not depend on the total duration of the sounds, which was varied independently of the presence or absence of inter-segment gaps in Experiment 3.

The results discussed so far were based on the decision model described within the section "Data Analysis". We also investigated a different decision model that was solely based on the sound level of the loudest segment within each trial. For the data of Experiment 1, the average predictive power of such a model in terms of the AUC was 0.72 (SD = 0.07, range 0.57–0.87) and thus significantly lower than for the model based on a weighted average of the segment levels. Furthermore, a model based on only the loudest segment would result in an exclusive weight on the amplified segments for the conditions in which the segments received a 15 dB amplification, because the sound level exceeded the sound level of all other segments on virtually all of the trials. However, as shown in Fig 4, most of the weights on the unamplified segments still differed from zero. For an attenuation of 15 dB, the "maximum" model would predict zero weights, which is indeed in line with our results. However, the "maximum" model would not predict the primacy effect on the unattenuated segments, as it was observed in Experiment 2, and also not a primacy effect in the baseline condition (as observed for the contiguous sounds).

## Is there an interaction between the primacy effect and loudness dominance?

One aim of the present study was to assess whether the primacy effect and the effect of relative level interact with one another. Therefore, in both Experiment 1 and Experiment 2, we varied the temporal position of the level change. In Experiment 1, sounds in which the segments were separated by 500-ms gaps were presented to investigate the effects of different positions of level change in the absence of a primacy effect. In contrast, in Experiment 2, contiguous sounds were presented to investigate the effects of different positions of level change in conditions in which a primacy effect occurred in the baseline condition. If the primacy effect and the effect of relative level showed some interaction, one would expect different effects of the position of the level change in Experiment 2 compared to Experiment 1. For amplification, two ways of an interaction would have been possible. First, the primacy effect could have further increased the effect of relative level and thus could have resulted in a larger amplification-induced change in the weight on the first segment compared to the changes in weights on the middle segments where the temporal weights were already substantially lower in the baseline condition. Alternatively, it could be the case that when the weight on the first temporal segment is already increased due to the primacy effect, an amplification of this segment does not result in a substantial further increase in the segment weight. For an attenuation, also two ways

of an interaction would have been possible. The change in the weight on the first segment could have been smaller compared to the changes in the weights on the middle segments because the primacy effect on the first segment could have compensated for some of the effects of relative level, whereas on the middle segments, the reduction in weight due to the effect of relative level would have taken place on temporal weights that in the baseline condition were already quite low. Alternatively, attenuation could exert a particularly stronger effect when the weight in the baseline condition is elevated due to the primacy effect.

It turned out that in the conditions with attenuation, there was a significant effect of the position of the level change in Experiment 2. The weight factor was larger when the first segment was attenuated, compared to when a middle segment was attenuated. However, this effect was unlikely to be an interaction of the primacy effect and loudness dominance, because the same effect was also observed in Experiment 1, where no primacy effect was present in the baseline condition. For amplification, in Experiment 2 (contiguous sounds) the effect of relative level resulted in smaller weight factors for the first segment compared to the third segment (see Table 4), whereas for Experiment 1 (sounds with gaps) this was not the case (see Table 2). Based on these results, we conclude that for attenuation, the primacy effect and loudness dominance show no substantial interaction, because the effects of the position of the level change did not differ substantially between sounds with and without gaps. For amplification, however, the data showed a pattern compatible with an interaction between the effect of relative level and the primacy effect, in the sense that the position of the level change had an effect only for the contiguous sounds presented in Experiment 2 (where a primacy effect was present in the baseline condition), but not for the sounds with gaps presented in Experiment 1.

## Loudness dominance, primacy effect and masking effects

As mentioned in the introduction, a relationship between loudness dominance and forward and backward masking effects on the auditory intensity resolution (e.g., [19,20]) has been suggested (e.g., [8,15,18]). Following this line of reasoning, temporal portions of a sound that have a higher mean level compared to other temporal parts of the stimulus should be more informative for the loudness judgment task, because the listeners should be more sensitive to the level changes imposed on these segments with higher mean level, compared to level changes imposed on segments with lower mean level at least if those segments are temporarily close to the segments with higher mean level. Such an effect would result in an assignment of higher weights on the temporal portions with higher mean level if listeners would follow an "ideal observer" strategy [15,21]. In the present study, we observed strong effects of relative level on the segment weights in contiguous sounds. However, pronounced effects of relative level on the temporal weights were also found for sounds containing segments separated by 500-ms gaps. This indicates that the loudness dominance effect cannot be entirely attributed to backward- or forward-masking effects on the intensity resolution, since with 500-ms silent gaps, the non-simultaneous masking effects by segments with a higher level on the neighboring segments with a lower level should have been small [19,22]. Still, a comparison between Experiment 1 and Experiment 2 showed that the effects of relative level on the weight factors were stronger for contiguous sounds in Experiment 2 compared to sounds that were separated by 500-ms gaps presented in Experiment 1. A comparison of three sound types in Experiment 3 in a within-subjects design largely confirmed this finding of stronger effects of the relative level on the weight factors for contiguous sounds. The relative effect of attenuating one segment was substantially larger for sounds with contiguous 475-ms and 100-ms segments compared to sounds with 100-ms segments separated by gaps. For amplification, the effects of sound type on the effects of relative level failed to reach statistical significance, but

descriptively, there was also a trend towards increased effects of relative level on the weight factors for the contiguous sounds. These patterns are compatible with the assumption that masking effects on intensity resolution enhance the loudness dominance effect. Non-simultaneous masking effects on the intensity resolution might have also contributed to the reduced weights (relative to the baseline condition) on the two segments neighboring the amplified segments, as observed in Experiment 2 and Experiment 3. This effect is compatible with a reduction in intensity resolution in the temporal vicinity of the amplified segment.

The asymmetry of the effects of amplification and attenuation on the weight factors is also in line with an explanation based on masking effects on the intensity resolution. When a single segment within the sound is amplified, it could exert masking effects on its neighboring segments. However, an attenuated segment could be affected by masking effects caused by *several* neighboring segments higher in level. This is compatible with the stronger effect of attenuation compared to amplification on the weight factors, which was observed in most conditions in our experiments. However, not all of the observed effects can be attributed to masking on the intensity resolution. Since non-simultaneous masking effects vanish over time [19,22], the effect on the temporal weights should be reduced or completely absent for temporal segments with sufficient temporal distance to the amplified segments. As a consequence, for example, an amplification of the first segment should have had only a minor masking effect on the temporal weight of the fourth segment. For the ten-segment sounds in Experiment 2, this would imply that the majority of the segments were hardly affected by the amplification of the first segment. Therefore, solely on the basis of masking effects on intensity resolution, it is unclear, why those segments still showed weights that were substantially lower than the weight on the amplified segment. Another prediction based on the masking explanation would be that level changes of the first segments should lead to a smaller effect of relative level on the temporal weights compared to a level change of a middle segment. For the latter, both forward and backward masking effects would occur for both amplification and attenuation, whereas for the former, only one masking type occurs (in case of an amplification only forward masking affecting the following segments, and in case of an attenuation only backward masking by the following segments, affecting the attenuated segment itself). For amplification, this is in accordance with the results from Experiment 2, while for attenuation, the pattern of results is not compatible with this assumption. Thus, the data from the three experiments only partially support the assumption of non-simultaneous masking in intensity resolution as the main reason for the observed effects of relative level.

The finding that the effect of relative level on the weight factors showed significant differences between contiguous and non-contiguous sounds only for attenuation may also be partly explained by auditory forward and backward masking in signal detection [36], i.e., a detection threshold that is higher than the threshold in quiet when a non-simultaneous masker is added. In the following, we refer to this type of masking as pre- and post-masking [37,38] to distinguish it from the forward and backward masking in intensity resolution. The unattenuated segment prior to the attenuated segment might be masking at least the first few milliseconds of the attenuated segment (acting as a post-masker), which could reduce the overall loudness of the attenuated segment. Dau et al. [39] used processing stages with time constants from 5 to 500 ms to predict post-masking, reflecting the fact that that masking in detection can last for more than 100 ms [38]. The effect of pre-masking is presumably not contributing to the loudness judgments, since the pre-masking only occurs over about 20 ms [38], i.e., over a considerably shorter period than post-masking (see also [40]). However, at least two results indicate that pre- and post-masking cannot account for the whole effect. First, the effect of an amplified segment should be stronger on the following segment than on the preceding segment since the post-masking effect should be stronger than the pre-masking effect. The opposite trend is

found in Experiment 3 (see Fig 6). Second, post-masking (and pre-masking) cannot explain the effects of relative level observed in Experiment 1, where there were 500-ms gaps between the segments.

The effects of forward masking in intensity resolution have also been suggested as a possible source of the primacy effect [8]. The loudness judgements for the sounds containing gaps between the segments in Experiment 1 and Experiment 3 did not show any primacy effects in the unattenuated/unamplified baseline condition. As besides the 500-ms gaps within the sounds, all of the properties of the sounds were identical to that of the contiguous unattenuated/unamplified four-segments baseline condition of Experiment 2 and Experiment 3 for which a small primacy effect was visible, we assume that the lack of a primacy effect is due to the short gaps within the sounds. Recent research from our lab suggests that gaps within a sound result in a recovery of the mechanism(s) that cause the primacy effect [8]. When sounds contain gaps of 500 ms between the segments, as in Experiment 1 and Experiment 3, it can be assumed that within each gap a substantial recovery of the mechanisms takes place, which leads to almost uniform weights in the baseline condition with a flat level profile. As discussed by Fischenich, Hots [8], the lack of a primacy effect for sounds containing longer gaps is compatible with forward masking effects on the intensity resolution, although there are alternative explanations like the response characteristics of the auditory nerve (AN) fibers [8,10,14], evidence integration processes [8,10], or attentional mechanisms [10,14]. Furthermore, one has to note that we did not measure the intensity resolution for specific segments in the present study, but based our reasoning on previous research concerning non-simultaneous masking effects on the intensity resolution. It would be an interesting approach for future experiments to explicitly measure the intensity resolution for "isolated" segments in suitable control conditions and to compare the measured intensity resolution for the different segments to the pattern of temporal weights, to evaluate to which extent the temporal weights are correlated with the intensity resolution.

## Conclusion

In the present study, the effects of attenuating or amplifying single temporal segments within a sound on temporal loudness weights were examined in three experiments. Consistently, attenuation led to a decrease in the weight on the attenuated segment, whereas amplification led to an increase in the weight on the amplified segment. The effects of the relative level were already present at an attenuation or amplification by 5 dB. Part of the loudness dominance effect may be due to backward or forward masking effects on intensity resolution or, to a smaller extent, pre- and post-masking effects on detection. However, the loudness dominance effect was observed for sounds separated by 500-ms silent gaps in Experiment 1 and Experiment 3. Thus, the observed loudness dominance effect cannot be attributed entirely to backward or forward masking effects. The effect of amplification and attenuation on the weight factors were not symmetrical, with attenuation resulting in stronger relative changes of the weights in comparison with amplification. For attenuation, the effect on the weight factors differed between the temporal positions of the level change (first segment versus a middle segment). This difference between the effect of loudness dominance for the first and for a middle segment was largely independent of whether or not a primacy effect occurred in the baseline condition. For amplification, however, there was an effect of the position of the amplified segments on the weight factors only for the contiguous sounds presented in Experiment 2, where a primacy effect was observed in the baseline condition, but not for the sound with gaps presented in Experiment 1, where no primacy effect was observed. Therefore, we conclude that for attenuation, the present study indicates no interaction between the primacy effect and

loudness dominance, whereas for amplification, the data are compatible with an interaction between the two effects.

## Acknowledgments

We are grateful to Celine Gebbert, Jonas Krämer, Christoph Keßler, Anna-Lena Spies and Sinah Neuwardt for their assistance in data collection.

## Author Contributions

**Conceptualization:** Alexander Fischenich, Jan Hots, Jesko Verhey, Daniel Oberfeld.

**Data curation:** Alexander Fischenich, Daniel Oberfeld.

**Formal analysis:** Alexander Fischenich, Daniel Oberfeld.

**Funding acquisition:** Jesko Verhey, Daniel Oberfeld.

**Investigation:** Alexander Fischenich, Julia Guldan, Daniel Oberfeld.

**Methodology:** Alexander Fischenich, Jan Hots, Jesko Verhey, Daniel Oberfeld.

**Project administration:** Daniel Oberfeld.

**Resources:** Julia Guldan, Daniel Oberfeld.

**Software:** Alexander Fischenich, Jan Hots, Jesko Verhey, Daniel Oberfeld.

**Supervision:** Alexander Fischenich, Daniel Oberfeld.

**Validation:** Alexander Fischenich, Daniel Oberfeld.

**Visualization:** Alexander Fischenich, Daniel Oberfeld.

**Writing – original draft:** Alexander Fischenich, Jan Hots, Jesko Verhey, Julia Guldan, Daniel Oberfeld.

**Writing – review & editing:** Alexander Fischenich, Jan Hots, Jesko Verhey, Julia Guldan, Daniel Oberfeld.

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
