## [Decision Letter · Decision Letter 0]

23 Jul 2021

PONE-D-21-12847

Temporal loudness weights: Primacy effects, loudness dominance and their interaction

PLOS ONE

Dear Dr. Fischenich,

Thank you for submitting your manuscript to PLOS ONE. After careful consideration, we feel that it has merit but does not fully meet PLOS ONE’s publication criteria as it currently stands. Therefore, we invite you to submit a revised version of the manuscript that addresses the points raised during the review process.

We look forward to receiving your revised manuscript.

Kind regards,

Michael Döllinger, Ph.D.

Academic Editor

PLOS ONE

Journal Requirements:

Additional Editor Comments (if provided):

Please especially clarify the major issues raised by reviewer1, eg improve structure of the manuscript.

Reviewers' comments:

Reviewer's Responses to Questions

**Comments to the Author**

1. Is the manuscript technically sound, and do the data support the conclusions?

Reviewer #1: Yes

Reviewer #2: Yes

2. Has the statistical analysis been performed appropriately and rigorously? 

Reviewer #1: I Don't Know

Reviewer #2: Yes

3. Have the authors made all data underlying the findings in their manuscript fully available?

Reviewer #1: No

Reviewer #2: Yes

4. Is the manuscript presented in an intelligible fashion and written in standard English?

Reviewer #1: No

Reviewer #2: No

5. Review Comments to the Author

Reviewer #1: Temporal loudness weights: Primacy effects, loudness dominance and their interaction

The paper describes three experiments on loudness perception in groups of 8-10 listeners. The experiments were similarly designed. The listeners had to judge on the loudness of a set of stimuli (wbn) where the level of one of the stimuli (first, third or seventh) was lowered or increased. Gap duration between the single stimuli and duration of the stimuli varied in the experiments. It was found that attenuation has less influence on loudness than amplification. The primacy effect seems to be the dominant effect.

In summary, the paper is mostly correct. The presented experiments are appropriate to testing for the hypotheses. However, the experiments should be better justified in the introduction. In the current version it looks like a potpourri without a strong connection. Conclusions are mostly justified by the results. However, writing style is not appropriate and the readability is very difficult and somewhat tedious.

I would like to encourage the authors to rewrite the manuscript and to provide a better guidance for the reader. For example the hypotheses should be presented clearer.

E.g. the figures illustrating the stimulus settings could be increased: Why do you show example levels? It would be more intuitive to provide mean values (lines and standard deviations ) “ The arbitrary bars can be omitted. , It is hard to understand help

Please justify the use of different mean levels. What is the meaning of the chosen values (56.125 dB SPL)?

I am not familiar with the concept of intensity resolution. However, I would expect another measurement to quantify intensity resolution.

What was the meaning of different definitions for weights of attenuation and amplification (Eq.2 and 3) Is this the caus for postive weights for attenuation?

Why do you explain ROC analysis? I am not sure whether this is rather necessary. D-prime and ROC-areas are very similar in their meaning. Maybe a scetch of the data analysis and a more extensive data representation (if necessary nin the appendix) would help.

I recommend a better description of the results and an extensive explanation of the derivation of the weights.

The writing style needs enhancement. For example, it is confusing when you write about Exp 2 (l303) and start with "Experiment 1 …."

Monster sentences like “For a level fluctuating sound, the probability that a given temporal segment …” should be avoided. It is quite hard to understand this .

Reviewer #2: General comment

The study by Fischenich and co-workers investigates loudness judgement of human listeners, in particular the primacy effect and its possible interaction with loudness dominance of temporal components within a sound. The authors report no such interaction for the attenuation of temporal sound components, whereas for amplification an interaction seems likely.

The study is well conducted, methods and statistics are sound, although some parts of the text are confusing (cf. below).

Specific comments

Major:

How the size of the effects of attenuation vs. amplification is described in the text is often confusing. For example:

Line 299: “the effect of attenuation was considerably larger than the effect of amplification” does not seem to reflect the figure, but it does fit to the table.

Another example: line 581f (“the effect of amplification was larger than the effect of attenuation”) seems to contradict line 701f (“…stronger effect of attenuation compared to amplification observed in the experiments”). To avoid such confusion, whenever such comparisons of effects of attenuation and amplification are made throughout the text, please describe exactly what effect is referred to.

Minor:

In the Data Analysis section the decision model is described (line 178ff) and the interpretation of the whole study is based on that model, but this is not discussed in the Discussion section. What impact would alternative models for loudness judgement, e.g. a rating based on the loudest segment only, have on the interpretation of the data? This should at least shortly be discussed.

Line 375: Remove extra full stop.

Line 396f: … with amplifying the beginning of the sound resulting in smaller weight factors compared to attenuation of the middle position.

Line 414f: With respect to Fig. 4, this seems to be true for the third segment, but not for the 7th.

6. PLOS authors have the option to publish the peer review history of their article (what does this mean?). If published, this will include your full peer review and any attached files.

Reviewer #1: No

Reviewer #2: No

---

## [Author Response · Author response to Decision Letter 0]

22 Oct 2021

We thank the reviewers for the detailed and helpful reviews and addressed each of the raised points within the revised manuscript and described the changes we made within the response letter uploaded as "Response_Fischenich_et_al_Loudness_Reviewer1_2". We hope that the manuscript now meets the expectations of the reviewers.

---

## [Decision Letter · Decision Letter 1]

23 Nov 2021

Temporal loudness weights: Primacy effects, loudness dominance and their interaction

PONE-D-21-12847R1

Dear Dr. Fischenich,

We’re pleased to inform you that your manuscript has been judged scientifically suitable for publication and will be formally accepted for publication once it meets all outstanding technical requirements.

Kind regards,

Michael Döllinger, Ph.D.

Academic Editor

PLOS ONE

Additional Editor Comments (optional):

Reviewers' comments:

Reviewer's Responses to Questions

**Comments to the Author**

1. If the authors have adequately addressed your comments raised in a previous round of review and you feel that this manuscript is now acceptable for publication, you may indicate that here to bypass the “Comments to the Author” section, enter your conflict of interest statement in the “Confidential to Editor” section, and submit your "Accept" recommendation.

Reviewer #1: All comments have been addressed

Reviewer #2: All comments have been addressed

2. Is the manuscript technically sound, and do the data support the conclusions?

Reviewer #1: Yes

Reviewer #2: Yes

3. Has the statistical analysis been performed appropriately and rigorously? 

Reviewer #1: Yes

Reviewer #2: Yes

4. Have the authors made all data underlying the findings in their manuscript fully available?

Reviewer #1: Yes

Reviewer #2: Yes

5. Is the manuscript presented in an intelligible fashion and written in standard English?

Reviewer #1: Yes

Reviewer #2: Yes

6. Review Comments to the Author

Reviewer #1: Thank you for the revision of the manuscript. All of my questions were answered and all of the issues raised were corrected or sufficiently justified.

Reviewer #2: The authors have made significant improvements to the manuscript. All my suggestions have been addressed to my satisfaction, I have no further comments.

7. PLOS authors have the option to publish the peer review history of their article (what does this mean?). If published, this will include your full peer review and any attached files.

Reviewer #1: No

Reviewer #2: No

---

## [Editor Report · Acceptance letter]

14 Dec 2021

PONE-D-21-12847R1 

Temporal loudness weights: Primacy effects, loudness dominance and their interaction 

Dear Dr. Fischenich:

I'm pleased to inform you that your manuscript has been deemed suitable for publication in PLOS ONE. Congratulations! Your manuscript is now with our production department. 

Kind regards, 

on behalf of

Dr. Michael Döllinger 

Academic Editor

PLOS ONE